# The Effects of a Short Virtual Reality Training Program on Dynamic Balance in Tennis Players

**DOI:** 10.3390/jfmk8040168

**Published:** 2023-12-18

**Authors:** Dario Novak, Filip Sinković, Zlatan Bilić, Petar Barbaros

**Affiliations:** Faculty of Kinesiology, University of Zagreb, 10000 Zagreb, Croatia; filip.sinkovic@kif.unizg.hr (F.S.); zlatan.bilic@kif.unizg.hr (Z.B.); petar.barbaros@kif.unizg.hr (P.B.)

**Keywords:** training intervention, Y Balance Test, stimulus, placebo, experimental group

## Abstract

This study aimed to analyze the effects of a 5 min virtual reality training program (VR) on dynamic balance in tennis players. Fifty-eight college tennis players (mean age 22.9 ± 3.1 years, weight 73.9 ± 10.8 kg, height 176.6 ± 8.4 cm) were allocated to either the control group (placebo) (CG, *n* = 20) or the right-handed experimental group (RTG, *n* = 20) and left-handed experimental group (LTG, *n* = 18), both of which took part in the 5 min VR intervention program. Pre-tests included anthropometric measures and the Y Balance Test (YBT), and the post-test was the Y Balance Test (YBT). Three YBT trials were performed in anterior (ANT), posterolateral (PL), and posteromedial (PM) directions. After the training intervention, in the right-handed experimental group (RTG), significant differences were observed for two variables: anterior reach (right foot) (*p* = 0.00) and posterior medial reach (right foot) (*p* = 0.03). In other analyzed variables, there were no significant differences. Additionally, the effect size was small. In the left-handed experimental group (LTG), statistically significant improvements were identified in five out of six analyzed variables: anterior reach (left foot) (*p* = 0.00), posterior medial reach (left foot) (*p* = 0.00), posterior lateral reach (left foot) (*p* = 0.00), posterior medial reach (right foot) (*p* = 0.00), and posterior lateral reach (right foot) (*p* = 0.00). The effect size ranged between small and medium. No significant changes were observed in the control group (CG) after the training intervention. Moreover, for all variables, the time*group interaction is determined. Anterior reach (left foot), posterior medial reach (left foot), posterior medial reach (right foot), and posterior lateral reach (right foot) showed significant interactions (F = 3.04, *p* = 0.05; F = 3.50, *p* = 0.03; F = 6.08, *p* = 0.00; and F = 4.69, *p* = 0.01). The outcome of this study leads us to a further understanding that if a player were to partake in VR activity, it could show a positive effect on their dynamic balance performance.

## 1. Introduction

Using virtual reality (VR) as a training tool is becoming increasingly popular. VR provides an immersive digital space where users can interact with objects and navigate as if they were present in the virtual environment, which presents as a real environment to the user [1,2]. VR technology has gained interest in situations where real-world training is logistically challenging to organize, dangerous, or impractical [2]. However, in specific application areas like VR sports training, there has been limited research to test this assumption. While the potential benefits of VR in sports training are promising, it is essential to conduct comprehensive and long-term studies to fully understand its effectiveness across various sports disciplines and different skill levels. Additionally, investigating the long-term retention of skills acquired through VR training and comparing them to traditional training methods may provide valuable insights into the technology’s true potential in the realm of sports [2]. As the sporting world embraces these innovative methods, it becomes essential for coaches and athletes to adapt and leverage VR technology as a valuable tool for reaching new levels of excellence.

Applying VR to sports training has several advantages. First, it offers the possibility for individuals to train without requiring access to the necessary sporting environment or multiple training partners. Second, actively integrating VR into sports training enables users to log their performance and closely monitor their progress. Third, VR is scalable and offers a high degree of freedom to create and control virtual environments in various ways. These advantages provide a significant opportunity to integrate evidence-based practices to achieve greater performance improvements from training [3,4]. Also, VR can transform practice sessions into engaging and immersive games, making training more enjoyable and motivating for athletes. 

Various sports have been explored in VR training studies, particularly table tennis [5,6,7,8]. One of the examples discovered that target accuracy significantly improved in the virtual training group compared to a group that received coaching from a table tennis professional. However, despite the virtual training group’s enhanced target accuracy, their overall technique deteriorated in comparison to the ideal movement strokes for this task [5]. Furthermore, one of the previous studies investigated the effect of VR technology on baseball training. A group that underwent repetitive real-world batting practice was compared to a group that engaged in repetitive batting practice within a virtual environment, as well as a group that underwent adaptive training within a virtual environment. The results showed that adaptive training led to a significant improvement in real-world performance compared to both groups that solely received repetitive practice [9].

Tennis competitions are characterized by a fast pace, high power, and multiple facets of tactics. Due to the fast-paced nature of the game, players must anticipate, move, adjust, and be prepared to hit the ball [10,11,12,13]. The ability to maintain balance and stability while executing these dynamic movements has garnered significant attention from two studies [12,13]. Dynamic balance, a crucial aspect in tennis, refers to the body’s ability to instinctively regulate its posture to maintain relative stability while in motion or when subjected to external forces [12]. Several factors influence dynamic balance, including the height of the center of gravity, the size of the support surface, and various stability factors. Achieving and maintaining dynamic balance depends on the coordination of visual, proprioceptive, and vestibular information in the central nervous system, as well as precise control of motor effectors. This skill serves as a fundamental indicator of a tennis player’s overall quality and performance on the court [13]. Moreover, dynamic balance in tennis is not merely about maintaining physical stability; it plays a pivotal role in the execution of effective strokes, rapid changes in direction, and swift court coverage. Tennis players must seamlessly shift their weight from one foot to the other during shots, such as forehands and backhands, and adapt to the ever-changing ball trajectory [13]. To achieve this, dynamic balance must be accompanied by coordinated footwork, precise timing, and skillful proprioceptive control [13]. The power of VR for tennis training goes beyond mere visual simulation. It caters to a player’s visual, auditory, and kinesthetic learning styles through providing a highly immersive experience that simulates specific activities from a first-person stereoscopic perspective. This simulation not only creates a profound sense of presence but also actively assists the player in developing perceptual and motor skills [14,15]. Given these considerations, there is a strong potential connection between VR technology and athletes’ dynamic balance. This suggests that the incorporation of VR into standard training protocols can offer tennis players a unique opportunity to enhance their abilities and potentially reduce the risk of injuries through improving their dynamic balance skills. Also, VR can create immersive training scenarios where players can practice dynamic balance while responding to virtual tennis balls. Additionally, it offers real-time feedback and performance tracking, allowing athletes and coaches to monitor progress and make necessary adjustments [14]. Through improving dynamic balance via targeted VR training, tennis players may enhance their on-court agility, precision, and overall performance. These skills are vital in modern tennis, where split-second reactions and seamless transitions between offensive and defensive movements can make the difference between winning and losing [15].

Due to these positive results and the previous lack of research in tennis populations, this study aimed to analyze the effects of a 5 min VR training program on dynamic balance in tennis players. It was hypothesized that tennis players who received the VR training program may demonstrate significantly better dynamic balance scores using the Y Balance Test (YBT).

## 2. Materials and Methods

### 2.1. Participants

Fifty-eight college tennis players (mean age 22.9 ± 3.1 years, weight 73.9 ± 10.8 kg, height 176.6 ± 8.4 cm) were allocated based on their dominant hand to either the control group (placebo) (both-handed) (CG, *n* = 20) or the dominant right-handed experimental group (RTG, *n* = 20) and dominant left-handed experimental group (LTG, *n* = 18), both of which received the 5 min VR intervention. The G-Power program (version 3.1.9.2; Heinrich Heine University, Dusseldorf, Germany) was used to estimate the appropriate number of participants, with an expected effect power of f = 0.33, an alpha level of 0.05, and a statistical power of 0.90. To participate in the study, all participants had to meet specific inclusion criteria, including being physically active college athletes who train at least three times per week and being intermediate or advanced club-level players, who participated in college games. Exclusion criteria included any injury that may affect physical activity and performance at the start of the study. Participants were also excluded from the study if they had a history of neurological disorders or medical conditions affecting proprioception or vestibular function. This additional criterion ensured that the participants’ balance and motor control were not compromised by underlying health conditions, thus preserving the integrity of the results recorded. The study was conducted in accordance with the Helsinki Declaration and approved by the Ethics Committee of the Faculty of Kinesiology, University of Zagreb (protocol code number 92; date of approval: 11 October 2023). All participants were informed of the purpose of the research and the conditions for participation. They also provided prior written consent to participate. The complete testing protocol was explained to them in detail, with a special emphasis on the additional effort required for the research and the risk of injury, which notably was at the same level as during standard training or competitions.

### 2.2. Measurements

Pre-tests included anthropometric measures and the Y Balance Test (YBT), while the post-test was the Y Balance Test (YBT). Body height (cm) was measured using a portable altimeter (Seca 213, Seca GmbH, Hamburg, Germany), while body mass (kg) was measured using a portable digital scale (Seca V/700, Seca GmbH, Hamburg, Germany). The Y Balance Test setup followed recommended standardization protocols, utilizing the Y Balance Test™ kit equipment (YBT Kit, Move2Perform, Evansville, IN, USA) [16]. The kit consists of a stance platform to which three pieces of PVC pipe are affixed, and aligned with the anterior, posteromedial, and posterolateral reach directions. The posterior pipes are positioned at 135 degrees from the anterior pipe, with a 45-degree separation between them. Each pipe is marked at 5 mm intervals to facilitate precise measurements [16]. A 5 min virtual reality training program was created using a specialized virtual reality device (ImproVR GmbH, Munich, Germany). The device retains a program that incorporates neuro-athletic, coordinative, and cognitive tasks into exercises that are quantifiable and traceable.

### 2.3. Study Design and Procedure

Before the testing session, all participants completed a standardized warm-up specific to tennis. The warm-up consisted of various activities, including light-intensity running covering a distance of 10 × 20 m. Following this, participants engaged in dynamic stretching exercises for a total duration of 15 min. These dynamic stretches involved lateral movements, skipping, jumping, and lunges, and concluded with four repetitions of sub-maximum acceleration. The warm-up was followed by the initial implementation of the Y Balance Test as a pre-test, then the 5 min virtual reality protocol, and the final performance of the Y Balance Test as a post-test. The rest interval after the VR protocol was 30 s, as this was the time required for the participant to return to the starting position and prepare for the final performance of the Y Balance Test. After the final performance test, all participants had an additional five-minute cool-down period, which consisted of light jogging and static stretching to gradually reduce the heart rate and promote muscular recovery. This structured warm-up and cool-down routine ensured that all participants were physically prepared for the study’s assessments and that their physical condition returned to baseline after the VR training protocol.

#### 2.3.1. Y Balance Test

The athlete stood on one leg on the central footplate with the farthest part of the foot at the starting line. While maintaining a single-leg stance, the athlete was asked to reach with the free limb in the anterior, posteromedial, and posterolateral directions about the stance foot. To enhance test reproducibility and establish a consistent testing protocol, a standard testing order was developed and employed. The athlete was instructed to push the distance indicator as far as possible in the direction under evaluation. The researcher closely monitored the athlete during testing and ensured that the indicator was not moved through kicking it or accelerating it at the end of the push. The maximum reach distance was recorded at the furthest point the foot reached on the proximal edge of the indicator and was measured to the half centimeter. The testing order consisted of three trials in which the athlete stood on their left foot and reached in the anterior direction (left anterior reach), followed by three trials standing on their right foot and reaching in the anterior direction. This process was then repeated for the posteromedial and posterolateral reach directions. After three trials in a specific reach direction, the raters were asked if they had observed at least one successful trial. If not, the athlete was instructed to perform an additional trial until a successful reach was achieved [16].

#### 2.3.2. Short Virtual Reality Training Program

The primary objective of the interventions utilizing the virtual reality apparatus was to augment the athletes’ reflexive stability. The protocol of the short virtual reality training program consisted of three exercises. The experimental group received a 5 min VR intervention using virtual glasses. The control group was also provided with a 5 min virtual reality intervention, during which they observed a solitary point that was in motion, moving up and down. The subject stands still, freely observing the space without participating in any task. They did not have any intervention that would require a reaction to the movement, unlike the experimental group.

Exercise: Enhancement of Peripheral Perception

The participant is in a standing position with his feet shoulder width apart. After he is set up with a pair of VR glasses, he is given instructions to perform the exercises. The participant looks at a circle with crosshairs located in the middle of the space and directs his central focus onto it. The circle with crosshairs changes color from red to green. The participant must consciously keep their attention on a peripheral focus because red and green spots appear in the space around the circle with crosshairs. If the circle with crosshairs is red, the participant must touch and remove the red spots with a movement of his hand and finger, if it is green, the participant must remove the green spots. Alternately, every few seconds the circle with crosshairs will change color. The result represents the number of points according to the number of spots removed.

Objective: Stimulation of the mesencephalon to bolster cranio-cervical stability. Activation of the mesencephalon contributes to the stabilization of both hemispheres. Peripheral awareness predominantly stimulates the neural circuits associated with attention and the ambient visual processing regions within the brainstem, particularly the mesencephalon. This activation is instrumental in enhancing cranio-cervical and ocular stability, primarily through the tractus tectospinalis, and in augmenting proprioceptive awareness.

2.Exercise: Gaze stability

The participant looks at the point in the middle of the space with one eye. If the participant’s dominant side of their body is the right side, the task is performed with the right eye, if the dominant side is their left side, the task is performed with the left eye. The task of the participant is to consciously follow the point that is moving up and down with their central focus without interruption.

Objective: Gaze stability augments the concentration mediated via circuits within the frontal lobe. Achieving gaze stabilization necessitates intricate oculomotor control, orchestrated via subcortical structures such as the cerebellum and specific regions within the brainstem. Consequently, gaze stabilization can ameliorate balance, coordination, and the alignment and stability of the vertebral column, influencing one’s postural dynamics during movement.

3.Exercise: Execution of horizontal and diagonal saccades and pursuits

The participant must track the dot that appears in the space and track its movement. Every few seconds a new dot appears that goes in a different direction from the previous one.

Objective: Horizontal and diagonal saccades and pursuits stimulate the brainstem and achieve side-specific modulation of muscular tonicity patterns. Voluntary ocular movements are precipitated by the frontal and parietal lobe regions and are regulated via the brainstem. Depending on the directionality of these eye movements, one can induce a more hemisphere-specific neural activation pattern.

### 2.4. Statistical Analysis

Data were processed in the program Statistica 14.0.1.25 (TIBCO Software Inc., Palo Alto, CA, USA) on the Windows operating system, and in Microsoft Excel 2016 (Palo Alto, CA, USA). The normality of the distribution was tested with the Shapiro–Wilk W test. Descriptive statistics were used to determine the basic parameters of test results for each group (CG and TG) in the initial and final testing phases (mean—x̄; standard deviation—SD). To analyze the difference between the initial and final testing within groups, a *t*-test for dependent samples was used. Additionally, a percentage change was calculated for all variables, along with the Cohen’s d coefficient as an indicator of effect size. Thresholds for effect size were statistically set to the following parameters: non-significant (<0.20), small (0.20–0.50), medium (0.50–0.80), and large (>0.80). Additionally, the general linear two-way mixed model analysis of variance (ANOVA) with repeated measures (group × time) was performed. The effect sizes were calculated and reported as the partial eta (ηp^2^). The level of statistical significance was set at *p* < 0.05 and the confidence interval was 95%.

## 3. Results

Table 1 shows the mean values and standard deviations of the analyzed variables in all the groups. It also presents the results of a *t*-test for dependent samples, which analyzed the difference between the initial and final testing after the conducted intervention, along with Cohen’s d coefficient as an indicator of effect size. As expected, the control group did not exhibit significant differences between the initial and final tests, and the effect size was found to be non-significant. However, significant differences between the initial and final testing following the VR intervention were observed in the right-handed experimental group (RTG) for two variables: anterior reach (right foot) and posterior medial reach (right foot). In other analyzed variables, there were no significant differences. Additionally, the effect size was small. Additionally, statistically significant improvements in test results were identified in the left-handed experimental group (LTG) in five out of six analyzed variables, specifically: anterior reach (left foot), posterior medial reach (left foot), posterior lateral reach (left foot), posterior medial reach (right foot), and posterior lateral reach (right foot). The effect size ranged between small and medium. Table 2 shows the results of the mixed model analysis of variance (ANOVA) with repeated measures (group × time). Anterior reach (left foot), posterior medial reach (left foot), posterior medial reach (right foot), and posterior lateral reach (right foot) showed significant interactions.

## 4. Discussion

The findings of this study lead us to a further understanding of the potential effects of a short virtual reality training program on short-term dynamic balance in tennis players. After the training intervention, the experimental group demonstrated significant improvements (*p* < 0.05) in most measured variables, while no significant changes were observed in the control group. Additionally, the effect size for the control group was non-significant, while in both experimental groups, it ranged between small and medium. Also, the calculated percentage change (%) in results was small, but this was expected given the short duration of the test. The % change in the experimental groups ranged from 1% to 4%. When the time*group interaction is applied, most measured variables showed significant interactions.

The outcome of this study leads us to a further understanding that if a player was to partake in VR activity, it could show a positive effect on their dynamic performance. However, it should be understood that many factors lead to a person’s performance improving, and that VR is not the sole predictor of this improvement. However, integrating VR into standard training routines could present tennis players with a potential opportunity to elevate their capabilities and potentially decrease the risk of injuries through refining their dynamic balance skills [14]. Through targeted VR training to enhance dynamic balance, tennis players might potentially improve their on-court agility, precision, and overall performance. These skills are crucial in contemporary tennis, where split-second reactions and seamless transitions between offensive and defensive movements can determine the outcome of a match [15]. Despite the mentioned positive effects of VR technology on motor abilities in tennis, there is limited research exploring the effect of VR technology on dynamic balance in tennis. One study, conducted on a sample of skilled male (*n* = 14) and female (*n* = 14) tennis players aged between 12 and 17 years, demonstrated the significant effect of a 10 min VR training protocol on the development of motor abilities, movement, and stroke performance [17]. Additionally, according to a previous study, VR technology shows promise in enhancing various tennis-related skills. Specifically, using VR technology during training sessions can have a positive effect on ball-catching and perception, which are crucial in sports like tennis or baseball [6]. The short-term changes that VR technology can achieve, based on the results of this research, may help players with a warm-up routine just before the start of training or a match. It is possible to increase the level of muscle activation needed for better balance. Given the short duration of VR technology intervention, it is not possible to prejudge that the positive effects of using VR glasses may also apply to long-term changes in dynamic balance.

In addition to tennis, previous studies have highlighted the potential benefits of VR technology in various sports training contexts. In line with that, various sports have been explored in VR training studies, particularly table tennis [5,6,7,8]. In a previous study on a sample of 63 table tennis players, significantly better results (*p* < 0.05) on tests of motor abilities were demonstrated in the VR experimental group compared to the placebo group [5]. Also, one study revealed significant performance enhancements in batting in baseball, effectively bridging the gap between virtual and real-world sports skills. A group that underwent repetitive batting practice in the real world was compared to a group that engaged in repetitive batting practice within a virtual environment, as well as a group that underwent adaptive training within a virtual environment. The results showed that adaptive training, tailored to the individual’s success level during practice, led to a significant improvement in real-world performance compared to both groups that solely received repetitive practice [9]. Another study offered an improved learning experience compared to traditional and video-based learning methods, with notable enhancements in knowledge acquisition and comprehension. The results showed that the learning experience through VR technology proved to be superior to traditional learning methods, such as videos [14]. Furthermore, a prior study involving 57 participants, divided into a VR training group (*n* = 29) and a no-training control group (*n* = 28), demonstrated that VR training significantly elevated real-world table tennis performance, emphasizing the potential of VR to enhance both quantitative and qualitative aspects of sports skills [8]. Some researchers have suggested that VR may be most valuable as a complementary training device for individuals with inherent knowledge and experience of the sport [6]. One of the previous studies found that participants expressed expectations of having enjoyable, engaging, and interactive learning experiences when using virtual reality [17]. It is crucial to continue exploring the technology’s full potential and its seamless integration into sports training methodologies. As VR technology continues to advance, its role in sports training is expected to expand, offering athletes innovative tools to enhance their skills and reach peak performance levels.

This study has certain limitations that should be considered. First, the motor tests were conducted with a sample of convenience participants under controlled conditions. Future research should focus on implementing tests with a larger number of participants to prevent potential selection bias. Second, this study was based on a short virtual training program. Future research could explore more extended training programs and their effect on dynamic balance, including the monitoring of long-term effects. Also, to better understand the mechanisms underlying balance improvement through virtual training, future studies could incorporate additional biomechanical and neurological analyses. Finally, future research should involve participants from various competition categories to obtain more precise data for enhanced practical and scientific applications. It can be suggested that the current study’s design may serve as a starting point for similar research in tennis. These findings also provide some practical insights for coaches to create a wide range of VR training scenarios aimed at enhancing players’ motor abilities, especially concerning balance [18].

## 5. Conclusions

The results of this research confirmed the hypothesis, and the findings of this study lead us to a further understanding of the potential effects of a short virtual reality training program on short-term dynamic balance in tennis players. Through improving dynamic balance via targeted VR training, tennis players may potentially enhance their on-court performance. In line with this, VR technology has introduced a new dimension to sports training, offering potential practical applications that could benefit athlete development. Also, this research opens the door for similar studies and improvements in that field, harnessing the possibilities offered by VR technology.

## Figures and Tables

**Table 1 jfmk-08-00168-t001:** Pre- and post-tests with % change and effect size.

	Control Group (*n* = 20)			Experimental Group (Right-Hand) (*n* = 20)			Experimental Group (Left-Hand) (*n* = 18)		
	Pre x̄ ± SD	Post x̄ ± SD	% of Change (ES)	Pre x̄ ± SD	Post x̄ ± SD	% of Change (ES)	Pre x̄ ± SD	Post x̄ ± SD	% of Change (ES)
ANT (L) (cm)	66.0 ± 6.1	65.9 ± 5.5	−0.2 (0.02)	67.7 ± 5.3	69.0 ± 6.4	1.9 (0.22)	63.6 ± 6.9	65.9 ± 6.9 *	3.6 (0.33)
PM (L) (cm)	104.6 ± 8.3	103.5 ± 7.4	−1.1 (0.14)	106.3 ± 7.4	107.9 ± 8.5	1.8 (0.20)	101.9 ± 6.2	104.3 ± 5.6 *	2.4 (0.41)
PL (L) (cm)	99.9 ± 8.6	100.6 ± 7.0	0.7 (0.11)	101.7 ± 7.9	102.5 ± 8.5	0.8 (0.10)	96.4 ± 6.8	99.4 ± 6.6 *	3.1 (0.44)
ANT (R) (cm)	66.6 ± 5.6	66.9 ± 4.2	0.5 (0.06)	67.2 ± 6.0	69.1 ± 6.6 *	2.8 (0.30)	63.6 ± 8.7	64.8 ± 7.4	1.9 (0.15)
PM (R) (cm)	104.3 ± 7.1	103.5 ± 6.4	−0.8 (0.12)	106.1 ± 8.2	107.6 ± 7.7 *	1.4 (0.20)	100.1 ± 4.8	103.8 ± 5.2 *	3.7 (0.74)
PL (R) (cm)	101.7 ± 6.8	101.8 ± 5.9	0.1 (0.02)	102.7 ± 7.7	104.1 ± 6.7	1.4 (0.20)	94.6 ± 7.7	98.7 ± 7.5 *	4.3 (0.54)

Legend: ES—effect size; ANT (L)—anterior reach (left foot); PM (L)—posterior medial reach (left foot); PL (L)—posterior lateral reach (left foot); ANT (R)—anterior reach (right foot); PM (R)—posterior medial reach (right foot); PL (R)—posterior lateral reach (right foot); *—significant differences between pre- and post-training values (*p* < 0.05).

**Table 2 jfmk-08-00168-t002:** Mixed model analysis of variance (ANOVA) with repeated measures (group × time).

	Control Group (*n* = 20)		Experimental Group (Right-Hand) (*n* = 20)		Experimental Group (Left-Hand) (*n* = 18)		Interaction TIME*GROUP		
	x̄ ± SD	x̄ ± SD	x̄ ± SD	x̄ ± SD	x̄ ± SD	x̄ ± SD	F	*p*	Partial ŋ^2^
ANT (L) (cm)	66.0 ± 6.1	65.9 ± 5.5	67.7 ± 5.3	69.0 ± 6.4	63.6 ± 6.9	65.9 ± 6.9	3.04	0.05 *	0.10
PM (L) (cm)	104.6 ± 8.3	103.5 ± 7.4	106.3 ± 7.4	107.9 ± 8.5	101.9 ± 6.2	104.3 ± 5.6	3.50	0.03 *	0.11
PL (L) (cm)	99.9 ± 8.6	100.6 ± 7.0	101.7 ± 7.9	102.5 ± 8.5	96.4 ± 6.8	99.4 ± 6.6	2.36	0.10	0.08
ANT (R) (cm)	66.6 ± 5.6	66.9 ± 4.2	67.2 ± 6.0	69.1 ± 6.6	63.6 ± 8.7	64.8 ± 7.4	1.45	0.24	0.05
PM (R) (cm)	104.3 ± 7.1	103.5 ± 6.4	106.1 ± 8.2	107.6 ± 7.7	100.1 ± 4.8	103.8 ± 5.2	6.08	0.00 *	0.18
PL (R) (cm)	101.7 ± 6.8	101.8 ± 5.9	102.7 ± 7.7	104.1 ± 6.7	94.6 ± 7.7	98.7 ± 7.5	4.69	0.01 *	0.14

Legend: F—f value; *p*—statistical significance; partial ŋ^2^—effect size; ANT (L)—anterior reach (left foot); PM (L)—posterior medial reach (left foot); PL (L)—posterior lateral reach (left foot); ANT (R)—anterior reach (right foot); PM (R)—posterior medial reach (right foot); PL (R)—posterior lateral reach (right foot); *—significant differences between pre- and post-training values (*p* < 0.05).

## Data Availability

Data available on request.

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
