# Peer review of "The Effects of a Short Virtual Reality Training Program on Dynamic Balance in Tennis Players"

_jfmk, 2023, doi:10.3390/jfmk8040168_

Round 1

Reviewer 1 Report

Comments and Suggestions for Authors

The intent of the manuscript (The effects of a Short Virtual-Reality Training Program on Dynamic Balance in Tennis Players) is to assess the effects of VR on impacting dynamic balance in tennis players. VR has rapidly grown in popularity and its influence on improving numerous skills and techniques is promising in sports competitions and other field, so the topic is intriguing. However, the overall data is largely scant in this scope of work. Additionally, the methodology and study design appear to be flawed. These issues, as well as other major concerns, are highlighted below:

*Subject demographics have been ignored in terms of gender, race/ethnicity, dominant playing hand/arm, etc. A table should be added which includes this.  Also, what "level" of competition are the participants in terms of tennis playing ability?  Are they amateur, club, and/or professional level?  Please add this pertinent information.

*Information regarding the control group (placebo) is also lacking.  What did the "placebo" consist of?  Also, based on the end-points measuring being pretest/posttest, what is the need for a placebo group to begin with?  The most critical evaluation would be the VR group, so please clarify the rationale as to why this was included within the study.

*It appears that the VR training program was only one 5-minute training session, which would likely have no, or at the very least, minimal long-term incremental positive effects.  Why was this type of regimen selected?  Also, why was not any potential long-term benefit analyzed from this VR session?  For VR to be effect in the cohort selected, the change would need to be of long-duration.  There also appears to be some ambiguity with the timeframe between when pretest and posttest was conducted, so please clarify.

*The explanation of the actual VR training program needs a tremendous amount of work.  In its current form, it is virtually impossible for other scientists to reproduce these results, due to it being far too vague.  To enhance rigor, please be more detailed within this section.

*The addition of Effect Size is appreciated, but it does not appear that the p-values are provided, even though it states within the statistical analyses section that this was report.  Please be more explicit with this information.

*Overall, the amount of data provided within the manuscript does not seem sufficient enough to be relevant to the scientific community, with merely including 2 Tables with 6 similar types of end-points.  Unfortunately, with this in mind, a de novo study would like be conducted, with including more high-quality, robust end-points.

Comments on the Quality of English Language

The use of English language within the text of the manuscript seems sufficient.

Author Response

Dear Editor,

We are pleased to resubmit for publication the revised version of our manuscript. We appreciate the constructive criticisms of the Editor and the reviewers. We have addressed each of their concerns as outlined below.

RESPONSES TO THE COMMENTS FROM REVIEWER 1:

Summary:
The intent of the manuscript (The effects of a Short Virtual-Reality Training Program on Dynamic Balance in Tennis Players) is to assess the effects of VR on impacting dynamic balance in tennis players. VR has rapidly grown in popularity and its influence on improving numerous skills and techniques is promising in sports competitions and other field, so the topic is intriguing. However, the overall data is largely scant in this scope of work. Additionally, the methodology and study design appear to be flawed. These issues, as well as other major concerns, are highlighted below.

 We thank the editor and the reviewers again for their helpful comments, which we feel have improved our manuscript. We hope that with these modifications, our paper can now be accepted for publication.

Specific Comments:

*Subject demographics have been ignored in terms of gender, race/ethnicity, dominant playing hand/arm, etc. A table should be added which includes this. Also, what "level" of competition are the participants in terms of tennis playing ability? Are they amateur, club, and/or professional level? Please add this pertinent information.

Done. We have added sentences about gender, dominant playing hand, and the level of players. All changes have been made and marked in the text in the "Participants" section.

*Information regarding the control group (placebo) is also lacking. What did the "placebo" consist of? Also, based on the end-points measuring being pretest/posttest, what is the need for a placebo group to begin with? The most critical evaluation would be the VR group, so please clarify the rationale as to why this was included within the study.

Thank you for pointing this out. More information about the control group and the “placebo” has been added to the text under the section "Study Design and Procedure." The control (placebo) group was also provided with a 5-minute virtual reality intervention, during which they observed a solitary point that was in motion, moving up and down. They did not have any intervention that would require a reaction of the movement, unlike the experimental group. The VR group is included in this study because it is the aim of the research to see how short-term intervention with VR technology affects balance.

*It appears that the VR training program was only one 5-minute training session, which would likely have no, or at the very least, minimal long-term incremental positive effects. Why was this type of regimen selected? 

Done. We did not claim that by short-term use of VR technology, we can influence long-term positive effects in the analyzed variables. To be clearer, we added a sentence in section „Discussion“.

*Why was not any potential long-term benefit analyzed from this VR session? For VR to be effect in the cohort selected, the change would need to be of long-duration.

We stated that in limitations with sentence: “Future research could explore more extended training programs and their impact on dynamic balance, including the monitoring of long-term effects.”

*There also appears to be some ambiguity with the timeframe between when pretest and posttest was conducted, so please clarify.

Done. We have changed sentence in section „Study design and procedure“.

*The explanation of the actual VR training program needs a tremendous amount of work. In its current form, it is virtually impossible for other scientists to reproduce these results, due to it being far too vague. To enhance rigor, please be more detailed within this section.

Revised as requested.

*The addition of Effect Size is appreciated, but it does not appear that the p-values are provided, even though it states within the statistical analyses section that this was report. Please be more explicit with this information.

Revised as requested.

*Overall, the amount of data provided within the manuscript does not seem sufficient enough to be relevant to the scientific community, with merely including 2 Tables with 6 similar types of end-points. Unfortunately, with this in mind, a de novo study would like be conducted, with including more high-quality, robust end-points.

Thank you for pointing this out. We agree with you that the statistical techniques are incomplete, so we have performed additional analyses. We have presented the results of the general linear two-way mixed model analysis of variance (ANOVA) with repeated measures (group × time). Additionally, the effect sizes are calculated and reported as the partial eta (ηp2).

We thank the editor and the reviewers again for their helpful comments, which we feel have improved our manuscript. We hope that with these modifications, our paper can now be accepted for publication.

Sincerely,

Dr Dario Novak

Reviewer 2 Report

Comments and Suggestions for Authors

Article Analysis

JFMK (ISSN 2411-5142)

Manuscript ID - jfmk-2709050

Title

The effects of a Short Virtual-Reality Training Program on Dynamic Balance in Tennis Players

Please accept my judgment positively and constructively.

The Introduction is well structured and addresses the essential aspects to frame the problem under study.

The method is well presented and the heating and cooling protocols are adequate.

The results, despite being well presented, could be clearer, as the methodology does not use a more robust statistical technique.

The discussion should be improved and repeats some paragraphs from the introduction.

The conclusions may eventually be different if a different and more robust statistical technique is used.

After finalizing the revision of the manuscript, I would like to make a few comments:

Abstract

1) If it is possible, some important results/numbers and p values should be reported into the abstract, with a new statistical analysis.

2) Methods

Although the statistical techniques used are correct, they are incomplete! it would be used more objectively, to read the results and conclusions mut be conducted for each dependent variable the a general linear two-way mixed model analysis of variance (ANOVA) with repeated measures (group × time).

To account for any side effects, a mixed-factors analysis of variance (ANOVA) must be performed (within-subjects: side, between subjects) for the outcomes assessed on the right and left side, which resulted significant from the between-groups comparison and the interaction (GROUP × TIME). The effect sizes must be calculated and reported as the partial eta (ηp2).

3) The discussion should be improved and repeats some paragraphs from the introduction.

4) The conclusions may eventually be different if a different and more robust statistical technique is used.

Author Response

Dear Editor,

We are pleased to resubmit for publication the revised version of our manuscript. We appreciate the constructive criticisms of the Editor and the reviewers. We have addressed each of their concerns as outlined below.

 RESPONSES TO THE COMMENTS FROM REVIEWER 2:

Summary:
The Introduction is well structured and addresses the essential aspects to frame the problem under study. The method is well presented and the heating and cooling protocols are adequate. The results, despite being well presented, could be clearer, as the methodology does not use a more robust statistical technique. The discussion should be improved and repeats some paragraphs from the introduction. The conclusions may eventually be different if a different and more robust statistical technique is used. After finalizing the revision of the manuscript, I would like to make a few comments.

We thank the editor and the reviewers again for their helpful comments, which we feel have improved our manuscript. We hope that with these modifications, our paper can now be accepted for publication.

Specific Comments:

1) Abstract

If it is possible, some important results/numbers and p values should be reported into the abstract, with a new statistical analysis.

Revised as requested.

 2) Methods

Although the statistical techniques used are correct, they are incomplete! it would be used more objectively, to read the results and conclusions mut be conducted for each dependent variable the a general linear two-way mixed model analysis of variance (ANOVA) with repeated measures (group × time).

Thank you for this suggestion. We agree with you that the statistical techniques are incomplete. As you suggested, we have performed the general linear two-way mixed model analysis of variance (ANOVA) with repeated measures (group × time). The effect sizes are calculated and reported as the partial eta (ηp2).

3) Discussion

The discussion should be improved and repeats some paragraphs from the introduction.

Thank you for pointing this out. The discussion has been improved.

4) Conclusion

The conclusions may eventually be different if a different and more robust statistical technique is used.

Revised as requested.

We thank the editor and the reviewers again for their helpful comments, which we feel have improved our manuscript. We hope that with these modifications, our paper can now be accepted for publication.

Sincerely,

Dr Dario Novak

Reviewer 3 Report

Comments and Suggestions for Authors

L8: you could shorten by deleting the range: …tennis players (mean age 22.9……

L17: so they weren’t actually different?  If that is the case you should just state that there were no other statistically significant relationships

L19: delete more specifically.  …analyzed variables: anterior reach…

L22: in the short term only.  Since this was only a pre- post- after 5 min.  There is no appreciation for if these changes persisted

L27: virtual reality is probably a lowercased term

L34: delete more

L41: First (not Firstly)

L43: Second

L44: Third

L48: VR instead of spelled out

L56: is that 3 groups or 2 groups, hard to tell. 

L59: how is this tailored, need elaboration to appreciate what that means

L62: Tennis competition….. (delete modern)

L63: what is strong rotation?  And facets, not faucets

L63: Subsequently line: this does not make sense.  Maybe: Due to the fast pace nature of the game players must anticipate….

L66: I wouldn’t quantify 2 studies as “numerous”.  I would quantify it as 2 studies

L79: Delete furthermore.  Furthermore express a continuation of thought.  You are starting a new topic in this sentence

L84: It is not evident yet.  L84-L89 are all speculative.  It should be written that way

L92: not can, may be able to.  This again needs to illustrate that there is the potential and it warrants testing

L97: …lack of research in tennis populations…

L99: it was hypothesized (not expected)

L99-100: ..it was hypothesized that tennis players who received the VR training program would demonstrated significantly better dynamic balance scores using the Y-Balance Test (YBT).

L101-104: delete.  One study can not prove anything, plus you are speculating things that would require additional studies beyond a 5 min intervention

L107: see early comment about how to edit

How were they allocated

Did the control group have both right and left participants?  I assume the right group had right hand dominate individuals and the left had left dominant?

L116: what does trainability mean?  Need to quantify.  Are these club players, college athletes, etc. 

L131: curious why anthropometrics would be pre and posttested for a 5 min intervention?  I would expect absolutely no changes

L136: The kit consists of…

L139: delete meticulously (not necessary)

L141: solution?  What does this mean?

L143: citations to support “scientifically validated techniques to enhance specific cognitive abilities essential for elite athletes”.  Seems like references are necessary

L182: This is not clear.  Earlier you had allocated people to groups.  Now you are also saying that they are stratified based on deficiency.  Several questions:

1)     Its not clear from your data that there was stratification.  When I think of stratification that means you have subgroups within each cohort (example: you might have 10 people with YBT scores that are symmetrical and 10 that are not).  Did that happen?

2)     If that happened – you need to detail what the cutoffs were for the stratification

3)     What does discernable lateralized deficiency mean?  You only tested ANT, PL, PM

L190: What was the exercise?  This is a description of what you were trying to “treat” but no exercise was described

L196 and 203: the same

L211: Data was processed…

Table 1: non-significant or not significant (not insignificant)

L234: it should be stated that there were no other differences.  Just because there were differences does not mean that they actually were different

Table 1 and 2: Suggest using standard YBT abbrev:  ANT, PL, PM

ES: you state a 90% CI, but it isn’t presented.  Also, why not a 95%

Table 1 and Table 2.  Posttest not Posttests.  There was only 1 posttest

L276: but it was not statistically significant so it may not actually be negative.  Need to delete this statement

L278: start a new paragraph

L280 and on: it would be nice to know what was improved specifically and what the time frame was for testing.  Was it 5 min or was it longer?

L287-289: you said “some findings”  that would suggest more than 1 study

L316: First

L317: sample of convenience

L319: what are “more comprehensive results”  - avoid nondescript phrases

And what would be higher performing individuals.  Quantify.  Both your sample and future samples

L320: Second

L342: in the short term

L342: “In line with this..”.  It has not shown successful results in skill acquisition and coaching.  It only showed YBT improved in the short term

L343: I don’t think you can say this either. 

L345: this has nothing to do with your study, it is speculative – delete

L345 – L 355: delete all.  This has nothing to do with the boundaries of your study

Comments on the Quality of English Language

There are some edits that are necessary.  I have tried to provide some examples and guidance

Author Response

Dear Editor,

We are pleased to resubmit for publication the revised version of our manuscript. We appreciate the constructive criticisms of the Editor and the reviewers. We have addressed each of their concerns as outlined below.

RESPONSES TO THE COMMENTS FROM REVIEWER 3:

Specific Comments:

L8: you could shorten by deleting the range: …tennis players (mean age 22.9…)

Done.

 L17: so they weren’t actually different? If that is the case you should just state that there were no other statistically significant relationships.

Thank you for this comment. They weren't actually different, so in the text, it has been added that in other analyzed variables, there were no significant differences.

L19: delete more specifically.  …analyzed variables: anterior reach…

Done.

L22: in the short term only.  Since this was only a pre- post- after 5 min.  There is no appreciation for if these changes persisted

Done.

L27: virtual reality is probably a lowercased term

Done.

L34: delete more

Done.

L41: First (not Firstly)

Done.

L43: Second

Done.

L44: Third

Done.

L48: VR instead of spelled out

Done.

L56: is that 3 groups or 2 groups, hard to tell. 

Done.

L59: how is this tailored, need elaboration to appreciate what that means

Done.

L62: Tennis competition….. (delete modern)

Done.

L63: what is strong rotation?  And facets, not faucets

Done.

L63: Subsequently line: this does not make sense.  Maybe: Due to the fast pace nature of the game players must anticipate….

Done.

L66: I wouldn’t quantify 2 studies as “numerous”.  I would quantify it as 2 studies

Done.

L79: Delete furthermore.  Furthermore express a continuation of thought.  You are starting a new topic in this sentence

Done.

L84: It is not evident yet.  L84-L89 are all speculative.  It should be written that way

Done.

L92: not can, may be able to.  This again needs to illustrate that there is the potential and it warrants testing

Done.

L97: …lack of research in tennis populations…

Done.

L99: it was hypothesized (not expected)

Done.

L99-100: ..it was hypothesized that tennis players who received the VR training program would demonstrated significantly better dynamic balance scores using the Y-Balance Test (YBT).

Done.

L101-104: delete.  One study can not prove anything, plus you are speculating things that would require additional studies beyond a 5 min intervention

Done.

L107: see early comment about how to edit

Done.

How were they allocated? Did the control group have both right and left participants?  I assume the right group had right hand dominate individuals and the left had left dominant?

Thank you for pointing this out. They were allocated to the experimental groups according to their dominant hand, but in the control group, participants were randomly allocated, including both right and left-handed individuals. All of this has been added to the manuscript text.

L116: what does trainability mean?  Need to quantify.  Are these club players, college athletes, etc. 

Thank you for this comment. The participants in this study are college athletes, and the changes you mentioned have been made in the text.

L131: curious why anthropometrics would be pre and posttested for a 5 min intervention?  I would expect absolutely no changes

Thank you for this comment. Pre-tests included anthropometric measures and the Y-Balance Test (YBT), while the post-test included only the Y-Balance Test (YBT).

L136: The kit consists of…

Done.

L139: delete meticulously (not necessary)

Done.

L141: solution?  What does this mean?

Thank you for pointing this out. It has been corrected.

L143: citations to support “scientifically validated techniques to enhance specific cognitive abilities essential for elite athletes”.  Seems like references are necessary

Thank you for pointing this out. It has been corrected.

L182: This is not clear.  Earlier you had allocated people to groups.  Now you are also saying that they are stratified based on deficiency.  Several questions:

Thank you for pointing this out. We recognized this was a flaw sentence. We have, accordingly, revised this part of the text. 

1)     Its not clear from your data that there was stratification.  When I think of stratification that means you have subgroups within each cohort (example: you might have 10 people with YBT scores that are symmetrical and 10 that are not).  Did that happen?

Fully agree with you. There was no stratification. College tennis players were allocated based on their dominant hand to either the control group, the dominant right-handed experimental group and dominant left-handed experimental group.

2)     If that happened – you need to detail what the cutoffs were for the stratification

Revised as requested.

3)     What does discernable lateralized deficiency mean?  You only tested ANT, PL, PM

Revised as requested. We do hope it is much more clear explanation in the text now. Thank you for pointing this out.

L190: What was the exercise?  This is a description of what you were trying to “treat” but no exercise was described.

Thank you for this comment. We have revised the exercise explanation.

L196 and 203: the same

Revised as requested.

L211: Data was processed…

Done.

Table 1: non-significant or not significant (not insignificant)

Done. Changed to non-significant.

L234: it should be stated that there were no other differences.  Just because there were differences does not mean that they actually were different.

Done. It has been changed that in other analyzed variables, there were no significant differences.

Table 1 and 2: Suggest using standard YBT abbrev:  ANT, PL, PM

Thank you for pointing this out. It has been corrected.

 ES: you state a 90% CI, but it isn’t presented.  Also, why not a 95%?

Thank you for pointing this out. It has been corrected.

Table 1 and Table 2.  Posttest not Posttests.  There was only 1 posttest

Done.

L276: but it was not statistically significant so it may not actually be negative.  Need to delete this statement

Thank you for pointing this out. Done.

L278: start a new paragraph.

Done.

L280 and on: it would be nice to know what was improved specifically and what the time frame was for testing.  Was it 5 min or was it longer?

Done. A more detailed description and the duration of the VR protocol in the studies have been added.

L287-289: you said “some findings”  that would suggest more than 1 study

Thank you for pointing this out. It has been corrected.

L316: First

Done.

L317: sample of convenience

Done.

L319: what are “more comprehensive results”  - avoid nondescript phrases. And what would be higher performing individuals.  Quantify.  Both your sample and future samples.

Thank you for pointing this out. It has been corrected.

L320: Second

Done.

L342: in the short term

Done.

L342: “In line with this..”.  It has not shown successful results in skill acquisition and coaching.  It only showed YBT improved in the short term

Thank you for this comment. It has been corrected.

L343: I don’t think you can say this either. 

Thank you for pointing this out. It has been corrected.

L345: this has nothing to do with your study, it is speculative – delete

Done.

L345 – L 355: delete all.  This has nothing to do with the boundaries of your study

Done.

We thank the editor and the reviewers again for their helpful comments, which we feel have improved our manuscript. We hope that with these modifications, our paper can now be accepted for publication.

Sincerely,

Dario Novak

Round 2

Reviewer 1 Report

Comments and Suggestions for Authors

The authors did a nice job with the revisions, though I still think that overall the study in general would have been far better with a better experimental design and stronger end-points.  If this request is not practical or feasible, then the softening of the tone regarding the results/conclusions and more objective data interpretation is strong suggested.

Comments on the Quality of English Language

Use of English language seems adequate.

Author Response

12 – December – 2023

Dear Editor,

We are pleased to resubmit for publication the revised version of our manuscript. We appreciate the constructive criticisms of the Editor and the reviewers. We have addressed each of their concerns as outlined below.

RESPONSES TO THE COMMENTS FROM REVIEWER 1:

The authors did a nice job with the revisions, though I still think that overall the study in general would have been far better with a better experimental design and stronger end-points.  If this request is not practical or feasible, then the softening of the tone regarding the results/conclusions and more objective data interpretation is strong suggested.

We thank the editor and the reviewers again for their helpful comments, which we feel have improved our manuscript. We hope that with these modifications, our paper can now be accepted for publication.

Sincerely,

Authors

Reviewer 2 Report

Comments and Suggestions for Authors

Article Analysis 

JFMK (ISSN 2411-5142)

Manuscript ID - jfmk-2709050

Title

The effects of a Short Virtual-Reality Training Program on Dynamic Balance in Tennis Players

The authores accepted my judgment positively and constructively, and changed some statistical protocols more robust, and the results were presented with greater clarity.

The discussion and the conclusions have been improved.

Author Response

12 – December – 2023

Dear Editor,

We are pleased to resubmit for publication the revised version of our manuscript. We appreciate the constructive criticisms of the Editor and the reviewers. We have addressed each of their concerns as outlined below.

RESPONSES TO THE COMMENTS FROM REVIEWER 2:

The authores accepted my judgment positively and constructively, and changed some statistical protocols more robust, and the results were presented with greater clarity. The discussion and the conclusions have been improved.

We thank the editor and the reviewers again for their helpful comments, which we feel have improved our manuscript. We hope that with these modifications, our paper can now be accepted for publication.

Sincerely,

Authors

Reviewer 3 Report

Comments and Suggestions for Authors

No additional comments

Comments on the Quality of English Language

Just review for grammar here and there

Author Response

12 – December – 2023

Dear Editor,

We are pleased to resubmit for publication the revised version of our manuscript. We appreciate the constructive criticisms of the Editor and the reviewers. We have addressed each of their concerns as outlined below.

RESPONSES TO THE COMMENTS FROM REVIEWER 3:

No additional comments. Just review for grammar here and there.

We thank the editor and the reviewers again for their helpful comments, which we feel have improved our manuscript. We hope that with these modifications, our paper can now be accepted for publication.

Sincerely,

Authors

Round 3

Reviewer 1 Report

Comments and Suggestions for Authors

Instead of completing a more rigorous study de novo that includes more robust end-points and then submitting a stronger manuscript, the authors have decided to stick with their current manuscript.  If this manuscript does proceed to publication, then the language should still be changed in terms of replacing terms like "would" and "will" to "may" since the current data does not necessarily support a positive benefit or outcome.

Comments on the Quality of English Language

Use of English language appears appropriate.

Author Response

15 – December – 2023

Dear Editor,

We are pleased to resubmit for publication the revised version of our manuscript. We appreciate the constructive criticisms of the Editor and the reviewers. We have addressed each of their concerns as outlined below.

RESPONSES TO THE COMMENTS FROM REVIEWER 1:

Instead of completing a more rigorous study de novo that includes more robust end-points and then submitting a stronger manuscript, the authors have decided to stick with their current manuscript.  If this manuscript does proceed to publication, then the language should still be changed in terms of replacing terms like "would" and "will" to "may" since the current data does not necessarily support a positive benefit or outcome.

We thank the editor and the reviewers again for their helpful comments, which we feel have improved our manuscript. The language has been changed in terms of replacing terms like "would" and "will" to "may". We hope that with these modifications, our paper can now be accepted for publication.

Sincerely,

Authors
